# Fumaric Acids Do Not Directly Influence Gene Expression of Neuroprotective Factors in Highly Purified Rodent Astrocytes

**DOI:** 10.3390/brainsci9090241

**Published:** 2019-09-19

**Authors:** Kaweh Pars, Marina Gingele, Jessica Kronenberg, Chittappen K Prajeeth, Thomas Skripuletz, Refik Pul, Roland Jacobs, Viktoria Gudi, Martin Stangel

**Affiliations:** 1Clinical Neuroimmunology and Neurochemistry, Department of Neurology, Hannover, Medical School, 30559 Hannover, Germany; 2Department of Neurology, European Medical School, University Oldenburg, 26129 Oldenburg, Germany; 3Department of Neurology, University Clinic Essen, 45147 Essen, Germany; 4Department of Clinical Immunology and Rheumatology, Hannover Medical School, 30559 Hannover, Germany; 5Center for Systems Neuroscience, University of Veterinary Medicine, 30559 Hannover, Germany

**Keywords:** glia, astrocytes, dimethylfumarate, monomethylfumarate, neuroprotection, growth factors

## Abstract

(1) Background: Dimethylfumarate (DMF) has been approved for the treatment of relapsing remitting multiple sclerosis. However, the mode of action of DMF and its assumed active primary metabolite monomethylfumarate (MMF) is still not fully understood. Former reports suggest a neuroprotective effect of DMF mediated via astrocytes by reducing pro-inflammatory activation of these glial cells. We investigated potential direct effects of DMF and MMF on neuroprotective factors like neurotrophic factors and growth factors in astrocytes to elucidate further possible mechanisms of the mode of action of fumaric acids; (2) Methods: highly purified cultures of primary rat astrocytes were pre-treated in vitro with DMF or MMF and incubated with lipopolysaccharides (LPS) or a mixture of interferon gamma (IFN-γ) plus interleukin 1 beta (IL-1β) in order to simulate an inflammatory environment. The gene expression of neuroprotective factors such as neurotrophic factors (nuclear factor E2-related factor 2 (NGF), brain-derived neurotrophic factor (BDNF), glial cell-derived neurotrophic factor (GDNF)) and growth factors (fibroblast growth factor 2 (FGF2), platelet-derived growth factor subunit A (PDGFa), ciliary neurotrophic factor (CNTF)) as well as cytokines (tumor necrosis factor alpha (TNFα), interleukin 6 (IL-6), IL-1β, inducible nitric oxide synthase (iNOS)) was examined by determining the transcription level with real-time quantitative polymerase chain reaction (qPCR); (3) Results: The stimulation of highly purified astrocytes with either LPS or cytokines changed the expression profile of growth factors and pro- inflammatory factors. However, the expression was not altered by either DMF nor MMF in unstimulated or stimulated astrocytes; (4) Conclusions: There was no direct influence of fumaric acids on neuroprotective factors in highly purified primary rat astrocytes. This suggests that the proposed potential neuroprotective effect of fumaric acid is not mediated by direct stimulation of neurotrophic factors in astrocytes but is rather mediated by other pathways or indirect mechanisms via other glial cells like microglia as previously demonstrated.

## 1. Introduction

Multiple sclerosis (MS) is a chronic disease of the central nervous system (CNS) and is characterized by neuroinflammation, demyelination, and neuronal degeneration [1,2]. It is a major cause of neurological disability in young adults [3]. Fumaric acids are known to modulate the immune system and have been used in psoriasis treatment for many years [4]. Many investigations examined immunomodulatory properties of dimethylfumarate (DMF), and it is reported that fumaric acid is protective for neurons and glial cells and thus DMF is considered neuroprotective [5,6,7]. Although the mode of action is not yet fully understood dimethylfumarate has been approved as a prophylactic therapy for treatment of relapsing remitting multiple sclerosis. In vitro investigations demonstrated that DMF is hydrolyzed to its assumed bioactive primary metabolite monomethylfumarate (MMF) [8,9,10,11]. Several investigations postulate a neuroprotective effect of DMF by inducing the nuclear factor E2-related factor 2 (Nrf2) pathway and thus reducing toxic-oxidative stress [12,13]. In antigen presenting cells (APCs) DMF stimulates type II dendritic cells (DC), which results in impaired secretion of pro-inflammatory interleukin (IL) 12 and IL-23 and increased production of the anti-inflammatory cytokine IL-10 [12]. Furthermore, MMF induces the secretion of tumor necrosis factor alpha (TNFα) and anti-inflammatory IL-10 and IL-1RA in peripheral blood mononuclear cells (PBMC) in vitro [14]. This indicates an inhibitory effect on inflammatory cells and a supporting impact on regulatory cells [15]. However, there are only a few experimental studies available that explain the role of DMF and MMF within the CNS. Historically, astrocytes have been classified as supporting cells of the mammalian CNS [16]. More functional properties of astrocytes gradually came into light and emphasize their important role and influence in different neurological disorders, such as Parkinson’s disease, Alzheimer’s disease, hepatic encephalopathy, and hypoxia/ischemia [17]. Thus, astrocytes reveal functional properties which bear the potential for therapeutic intervention [18]. 

Recent studies demonstrated a key role of astrocytes in wound healing and repair [19] and in the regulation of de- and remyelination in the CNS [20]. Therefore, it is of interest to investigate the influence of fumaric acids on astrocytes. There are reports on an anti-inflammatory effect of DMF on astrocytes by inhibiting pro-inflammatory mediators such as inducible nitric oxide synthase (iNOS), TNFα, IL-1β, and IL-6 [21]. Recently we demonstrated that fumaric acids directly influence gene expression of neuroprotective factors in rodent microglia [5]. To further elucidate the mode of action, we tested the hypothesis that DMF and MMF modulate the production of neurotrophic factors and growth factors in highly purified astrocytes in vitro in the absence of microglia. We thus analyzed the gene expression of nerve growth factor (NGF), brain-derived neurotrophic factor (BDNF), glial cell-derived neurotrophic factor (GDNF), platelet-derived growth factor subunit A (PDGFa), fibroblast growth factor 2 (FGF2) and ciliary neurotrophic factor (CNTF) under the influence of DMF and MMF. Furthermore, we examined different time kinetics of pre-treatment with fumaric acids and different types of lipopolysaccharides (LPS), cytokines, and exposition protocols as stimulators of astrocytes in order to simulate a suitable inflammatory situation in the CNS.

## 2. Materials and Methods

### 2.1. Preparation and Culture of Highly Purified Astrocytes

Primary mixed glial cell cultures were prepared from neonatal Sprague-Dawley rats (P0–P3) as described previously [22]. Animals were maintained in the Central Animal Facility of the Hannover Medical School (MHH). All procedures were performed in compliance with the international guidelines on animal care and the review board for the care of animal subjects of the district government (Lower Saxony, Germany. Number: 2012/13). Brains were freed from meninges, the cerebellum, and the brain stem. Afterwards they were minced and further enzymatically dissociated by 0.1% trypsin (Biochrom, Berlin, Germany) and 0.25% DNase (Roche, Mannheim, Germany). The cells were then plated into culture flasks pre-coated with poly-l-lysine (PLL; Sigma-Aldrich, Hamburg, Germany). The flasks were filled up with medium consisting of Dulbecco´s Modified Eagle Medium (DMEM; Life Technologies Carlsbad, CA, USA), 1% penicillin/streptomycin (Sigma-Aldrich) and 10% fetal bovine serum (FBS; Biochrom, Berlin, Germany). Cultures were incubated at 37 °C and 5% CO_2_.

Microglial cells were removed on day 7 by shaking at 37 °C for 45 min at 180 rpm on an orbital shaker (Edmund Bühler, Heching, Germany) and afterwards the medium was replaced. After resting for at least 2 h, oligodendrocytes were removed by shaking at 37 °C at 180 rpm for 16–20 h. Supernatants, including oligodendrocyte precursor cells, were then removed and medium was replaced. One-hundred µM of antimitotic arabinosylcytosine (Ara-C; Sigma-Aldrich) was added to each flask. Medium containing Ara-C was removed after 72 h, and the cells were washed with phosphate-buffered saline and harvested in trypsin/EDTA (0.05/0.02%) solution (Biochrom). After counting, 3.0 × 10^5^ cells were plated into 6-well plates. Astrocytes obtained following this protocol were referred to as highly enriched with a purity of ~99% as judged by glial fibrillary acidic protein (GFAP) immunostaining as demonstrated previously [23,24].

DMF is approved as a prophylactic therapy of relapsing remitting multiple sclerosis. To simulate a comparable environmental situation after a resting time of at least 4 days the cells were pre-treated with 10 µM dimethylfumarate solution (DMF; Sigma-Aldrich) or 10 µM monomethylfumarate solution (MMF; Sigma-Aldrich) for 30 min or 24 h. In the control cultures the medium without MMF or DMF was changed accordingly. After 30 min or 24 h either 10 ng/mL lipopolysaccharide from *Escherichia coli* 055:B5 (LPS-E; Sigma-Aldrich), 100 ng/mL LPS-E, 10 ng/mL lipopolysaccharide from *Salmonella typhimurium* (LPS-S; Sigma-Aldrich), 100 ng/mL LPS-S, a cytokine mixture of 50 ng/mL interferon gamma (IFN-γ; PeproTech, Rocky Hill, NJ, USA) and 10 ng/mL IL-1β (PeproTech), or medium were added. After 3, 6, 12, 24, and 48 h, the supernatants and cells were collected.

There were no microscopical abnormalities nor suspicious changes of the cell cycle in the cultures so we did not assume a mycoplasma testing as indicated.

### 2.2. RNA Isolation and Reverse Transcription Polymerase Chain Reaction (RT-PCR)

Real-time quantitative polymerase chain reaction (qPCR) was performed for the genes *NGF*, *BDNF*, *GDNF*, *PDGFa*, *FGF2*, and *CNTF*. Ribonucleic acid (RNA) was extracted using the RNeasy Mini Kit (Qiagen, Hilden, Germany) according to the manufacturer’s instructions. RNA concentration was measured with a NanoDrop 2000 spectrophotometer (Thermo Fisher Scientific, Waltham, MA, USA). Complementary deoxyribonucleic acid (cDNA) was synthesized using the High Capacity cDNA Reverse Transcription Kit (Applied Biosystems, Foster City, CA, USA). For qPCR analysis, the StepOne^TM^ Real-Time PCR System and appropriate TagMan assay (Applied Biosystems, Waltham, MA, USA) were used (Table 1). The ΔΔCT method was used to determine differences in the expression between untreated and treated astrocytes. Gene expression was internally normalized against the housekeeping gene hypoxanthine-guanine-phosphoribosyl-transferase 1 (HPRT-1).

### 2.3. Statistical Analysis

All experiments were performed at least three times. GraphPad Prism version 5.02 was used for statistical analysis (GraphPad Software, Inc., La Jolla, CA, USA). One-way ANOVA (analysis of variance) followed by the Tukey’s multiple comparison test, or Bonferroni’s multiple comparison test for post hoc comparison was used for statistical analysis. Values are presented as the arithmetic means ± standard error of the mean (SEM). *p* < 0.05 was considered to indicate a statistically significant difference.

## 3. Results

### 3.1. DMF is Biologically Active and DMF and MMF are not Toxic in Vitro

First, we investigated a possible toxic effect of DMF and MMF on astrocytes in vitro. After an incubation of 24 h, 48 h, 72 h, and 96 h neither DMF (10 µM) nor MMF (10 µM) showed toxic effects on astrocytes in vitro. It is well described that DMF reduces T-cell counts in vivo and that DMF induces apoptosis of peripheral mononuclear blood cells (PBMC) in vitro [25,26,27,28]. Therefore, we investigated effects of DMF (10 µM) on PBMC in our cell culture conditions and could demonstrate that DMF is biologically active in vitro (Appendix A) in the applied concentration that is also thought to be relevant in vivo.

### 3.2. DMF and MMF have No Effect on Growth Factor Gene Expression in Highly Purified Activated Astrocytes

The expression of the growth factors NGF, BDNF, GNDF, PDGFa, FGF2, and CNTF was measured in astrocytes after exposure to DMF (10 µM) and MMF (10 µM) for various timepoints. Except for an upregulation of FGF2 gene expression after 12 h of DMF treatment both DMF and MMF had no effect on treated cells compared to control (Figure 1). Thus, DMF and MMF did not modulate the expression of the tested growth factors in unstimulated/resting astrocytes. In order to simulate an inflammatory environment as it may occur during an MS attack astrocytes were stimulated with a mixture of cytokines (50 ng/mL IFN-γ and 10 ng/mL IL-1β) that are known to activate astrocytes [29]. Because in patients DMF treatment is given continuously even before onset an MS attack the cells were pre-treated with DMF (10 µM) or MMF (10 µM) for 24 h and were then activated by the cytokine mix. Astrocytes were harvested 3, 12, 24, and 48 h after activation (Figure 2).

After cytokine stimulation gene expression of NGF and CNTF was significantly downregulated whereas gene expression of GDNF and PDGFa was increased. BDNF and FGF2 showed no measurable alterations. However, pre-treatment with DMF or MMF for 24 h had no influence on any of these activation-mediated regulations.

### 3.3. DMF and MMF have No Effect on Growth Factor and Cytokine Expression in Lipopolysaccharide Stimulated Astrocytes

In order to investigate another well-established inflammatory stimulus for astrocytes we used lipopolysaccharides (LPS) as a ligand of the Toll-like receptor 4 (TLR4) [30]. In a previous publication LPS from *Salmonella typhimurium* was used to describe effects of DMF on microglia and astrocytes [21]. However, since LPS from different bacteria show a structural diversity [31] and microglial TLR4 can differentiate between the class of LPS structures [32], we first wanted to determine potential effects of LPS from different bacteria sources. Hence, we investigated two different concentrations of LPS-E (lipopolysaccharide from *Escherichia coli* 055:B5) and LPS-S (lipopolysaccharide from *Salmonella typhimurium*) on cytokine, neurotrophic factor, and growth factor expression in astrocytes. Astrocytes were stimulated for 6 h with 10 ng/mL or 100 ng/mL of either LPS-E or LPS-S. Both LPS-E and LPS-S treatment led to a significant increase of gene expression of the pro-inflammatory mediator IL-1β, while there was a downregulated gene expression of the anti-inflammatory insulin-like growth factor 1 (IGF-1) and no change in FGF2 expression (Figure 3). There were no differences between different LPS sources and concentrations so we decided to use LPS-E for further investigations.

In analogy to the cytokine stimulation of astrocytes, we pre-treated primary rat astrocytes with DMF (10 µM) or MMF (10 µM) for 30 min and 24 h, respectively. This was followed by a stimulation with LPS-E (10 ng/mL) for another 6 h. Afterwards we studied the expression of TNFα, IL-6, IL-1β, iNOS, FGF2, PDGFa, and CNTF by qPCR (Figure 4). Although LPS had a strong effect on the gene expression in astrocytes, there were no effects of DMF or MMF detectable at both pre-treatment periods on the expression of the factors mentioned above.

## 4. Discussion

Multiple sclerosis lesions show a heterogeneity [33]. There are different mechanisms postulated to be involved leading to tissue damage in MS. Lassmann and colleagues described interindividual differences in mechanisms of tissue injury in MS lesions [1]. In some patients as a result of cytotoxic T lymphocytes, and macrophages releasing toxic products there is a chronic inflammatory process leading to myelin and axonal destruction [34,35], whereas in other patients an accumulation of immunoglobulins and components of activated complement indicate pathogenic antibodies [36,37]. Previous reports pointed out that some MS lesions show a characteristic oligodendrogliopathy with oligodendrocyte apoptosis [33,38] similar to early stages of white matter ischemia [39]. As one of the major insults in MS hypoxia is associated with raised production of oxygen and nitric oxide radicals, inducing an impairment of mitochondrial function with subsequent histotoxic hypoxia [1,40]. Hypoxia and inflammation are linked by the enzyme prolylhydroxylase (PHD) [41] which are responsible for breaking down hypoxia inducible factor 1 alpha (HIF-1α) [42]. The PHD-related pathways have an impact on nuclear factor kappa B (NF-κB), which influences the production of pro-inflammatory cytokines [43]. Hypoxia inhibits PHD, allowing HIF-1α and NF-κB become active, and intensify the inflammation, which can damage the vascular endothelium and promote the influx of leukocytes [40]. 

The beneficial influence of DMF is evident, and thus, the drug was approved for treating relapsing remitting MS as a prophylactic therapy to influence inflammation. The purpose of the present study was to further elucidate the mode of action of DMF and MMF on astrocytes. Hence, we investigated the influence of DMF and MMF on the gene expression of cytokines, growth factors, and neurotrophic factors in primary rat astrocytes. We used the most likely in vivo concentration after oral intake of DMF (10 µM) [21,44] and for MMF a comparable concentration to the maximal MMF concentration detected in serum of healthy subjects (10 µM) [8]. Previous publications described that DMF induces apoptosis of peripheral blood mononuclear cells (PBMC) in vitro [26,45]. As a proof of principal we used a similar concentration on PBMC that also showed an effect on the inhibition of PBMC proliferation comparable to published data (Appendix A). For a suitable simulation of a representative environment, e.g. an MS relapse during the treatment with DMF, we pre-treated the cells with DMF or MMF for 24 h and then stimulated them with the cytokines IFN-γ (50 ng/mL) and IL-1β (10 ng/mL) for different periods of time (3, 12, 24, and 48 h). Analysis of qPCR demonstrated that after cytokine activation expression of GDNF and PDGFa was upregulated while expression of NGF and CNTF was downregulated. There were no changes in the expression profiles of BDNF and FGF2. However, for all factors there was no further modulation by DMF or MMF. Thus, all measured alterations can be ascribed to the cytokine stimulation, irrespective of DMF or MMF pre-treatment.

We further investigated the influence of bacteria-derived LPS to simulate inflammation since previous reports have illustrated that DMF decreases the synthesis of pro-inflammatory factors such as TNFα, IL-6, IL-1 β, and iNOS in LPS-stimulated astrocytes [21]. LPS is widely used to simulate an inflammation on target cells [46]. In previous investigations, LPS from different bacterial sources was used and in different concentrations [47,48]. In order to determine possible concentration- and/or bacteria-dependent effects we compared the gene expression of cytokines and growth factors in astrocytes treated with two different concentrations, 10 ng/mL and 100 ng/mL, of two different types of LPS derived from *Escherichia coli* 055:B5 and from *Salmonella typhimurium*. In all approaches, the results showed an increased expression of the pro-inflammatory mediator IL-1β as well as a downregulation of the anti-inflammatory factor IGF-1 compared to control. The expression of FGF2 was unchanged in all experimental groups. Thus, the effects of LPS in different concentrations derived from different bacteria seem to be comparable when applied to highly purified astrocytes without any other stimulus.

In previous studies, different protocols of pre-treatment with DMF were used, from no pre-treatment before simulated inflammation [49] to 30 min of pre-treatment. We investigated two different periods of pre-treatment, 30 min and 24 h, before stimulation with LPS for another 6 h. Gene expression of cytokines, neurotrophic factors, and growth factors were analyzed by qPCR and demonstrated no effects apart from the LPS-induced changes. 

Recently we could demonstrate a direct influence of fumaric acids on gene expression of neuroprotective factors in microglia, like IGF-1 [5]. Here, we were not able to reproduce the effects described by Wilms et al. that DMF pre-treated astrocytes stimulated by LPS induce a significant downregulation of mRNA synthesis for IL-1β, IL-6, and TNFα, and a moderate reduction of mRNA synthesis for iNOS in astrocytes [21]. In our experiments, there was a significant increase after LPS stimulation but DMF and MMF had no influence on either pro-inflammatory factors, neurotrophic factors, or on growth factors. Applying the same statistical analysis as Wilms et al., we set the results of not pre-treated LPS stimulated cells as 100%, but this did not change our findings. These divergent results could be due to different protocols for the preparation of the cells. Since we put extra attention on the purity of our astrocyte cultures, we achieved the lowest possible microglia contamination that could either directly or indirectly affect the results.

Although DMF has been proven as an effective oral MS therapy, the immunomodulatory influence of the drug is still not fully understood and arguments need perpetual adjustments to new findings. Exemplary, former investigations claimed nuclear factor E2-related factor 2 (Nrf2) as the pivotal pathway for a possible neuroprotective action of DMF [6]. However, recent studies with Nrf2-deficient (Nrf2^−/−^) mice demonstrated a new perspective on the impotence of Nrf2 [50] as oral DMF uptake revealed similar effects in Nrf2^−/−^ and wild-type mice [51]. Similarly, DMF treatment of mice that were experimentally demyelinated did not lead to a protection of oligodendrocytes [26]. DMF has been confirmed to have an impact on different types of cells including T cells as one main target. Studies demonstrated that under oral therapy with DMF CD4+ cells including pro-inflammatory Th1 cells as well as CD8+ T cells are reduced whereas Th2 cells are increased [28,52,53,54]. Th1 cells activate astrocytes and microglia via pro-inflammatory cytokines, and thus induce myelin phagocytosis [24,55]. The effect of DMF on astrocytes is most likely indirectly mediated by reduction of Th1 cells, followed by reduced astrocyte activation. Hence, beside the mentioned indirect involvement of astrocytes there might be no direct influence of DMF or MMF on neurotrophic factors and growth factors in highly purified astrocytes as an additional mode of action of fumaric acids. In previous investigations, MMF induced in vitro effects only at higher concentrations than found in serum of patients after intake of 120 mg DMF (one Fumaderm^®^ tablet) [44]. The MMF concentration used in this study might have been too low to induce effects on astrocytes in vitro, but was the highest realistic in vivo concentration in the CNS.

The precise mode of action of DMF as an immunomodulating MS therapy still is unknown. To uncover potential specific effects on peripheral blood cells, CNS cells, and possible interactions, further research should be performed in future.

## 5. Limitations of the Work

There are some limitations of this work. In previous reports, a discrepancy in gene and protein levels has been described [56,57]. We could not demonstrate any influence of DMF or MMF on the mRNA level. However, for a more precise insight of the influence of DMF or MMF on astrocytes additional protein expression profiles should be supplemented in the future.

Previous publications described that rodent astrocytes react in a different way to toll-like receptor ligands or cytokines compared to human astrocytes [58]. We therefore cannot exclude that there is a possible direct effect of DMF or MMF on human astrocytes. However, previous studies have described concordant effects of DMF and MMF on both human and rodent astrocytes [59].

## 6. Conclusions

In summary, our results indicate that neither DMF nor MMF directly affect gene expression levels of pro-inflammatory factors, neurotrophic factors, and growth factors in highly purified astrocyte cultures, irrespective of the time of pre-treatment with fumaric acids or the mechanism of astrocyte stimulation.

## Figures and Tables

**Figure 1 brainsci-09-00241-f001:**
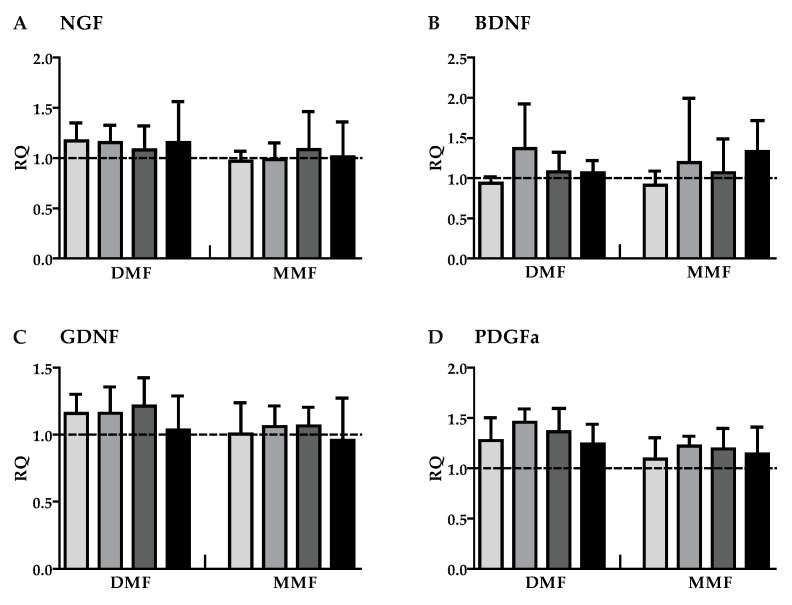
Gene expression of neurotrophic factors and growth factors in astrocytes treated with dimethylfumarate (DMF) or monomethylfumarate (MMF). Astrocytes were treated with medium, 10 µM DMF or 10 µM MMF for 24 h. Graphs show mRNA expression fold changes of NGF (**A**), BDNF (**B**), GDNF (**C**), PDGFa (**D**), FGF2 (**E**), and CNTF (**F**) after 3, 12, 24 or 48 h compared to the control group (astrocytes only treated with medium) and normalized with HPRT-1 using the ΔΔCT method. Data are presented as the arithmetic means ± SEM of *n* = 3–6. Significant differences are marked by asterisks (* *p* < 0.05; ** *p* < 0.01; *** *p* < 0.001).

**Figure 2 brainsci-09-00241-f002:**
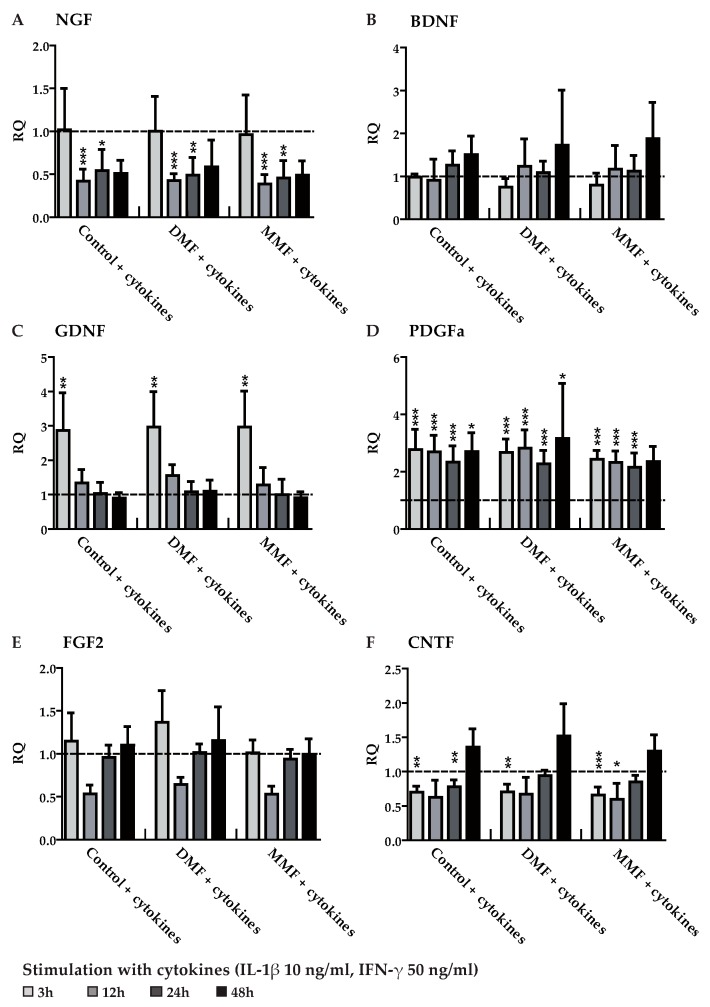
Gene expression of neurotrophic factors and growth factors in astrocytes pre-treated with DMF or MMF and stimulated with cytokines. Astrocytes were pre-treated with medium, 10 µM DMF or 10 µM MMF for 24 h and afterwards stimulated with cytokines (50 ng/mL interferon gamma (IFN-γ) and 10 ng/mL interleukin 1 beta (IL-1β)) for another 3, 12, 24 or 48 h. Graphs show mRNA expression fold changes of NGF (**A**), BDNF (**B**), GDNF (**C**), PDGFa (**D**), FGF2 (**E**), and CNTF (**F**) compared to the control group (astrocytes only treated with medium) and normalized with HPRT-1 using the ΔΔCT method. Data are presented as the arithmetic means ± SEM of *n* = 3–6. Significant differences are marked by asterisks (* *p* < 0.05; ** *p* < 0.01; *** *p* < 0.001).

**Figure 3 brainsci-09-00241-f003:**
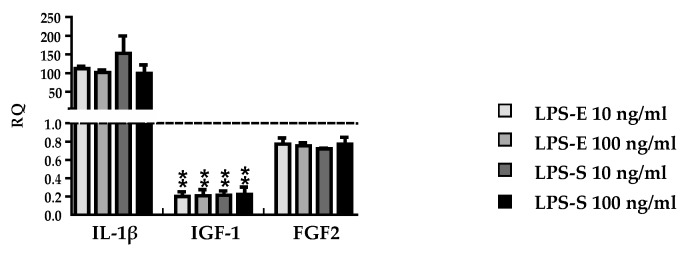
Effect of different types and concentrations of lipopolysaccharides (LPS) on gene expression of pro-inflammatory IL-1β and growth factors IGF-1 and FGF2. Astrocytes were stimulated for 6 h with 10 ng/mL or 100 ng/mL of either LPS-E (lipopolysaccharide from *Escherichia coli* 055:B5) or LPS-S (lipopolysaccharide from *Salmonella typhimurium*). Graphs show mRNA expression fold changes of IL-1β, IGF-1, and FGF2 compared to the control group (astrocytes only treated with medium) and normalized with HPRT-1 using the ΔΔCT method. Data are presented as the arithmetic means ± SEM of *n* = 4. Significant differences are marked by asterisks (* *p* < 0.05; ** *p* < 0.01; *** *p* < 0.001).

**Figure 4 brainsci-09-00241-f004:**
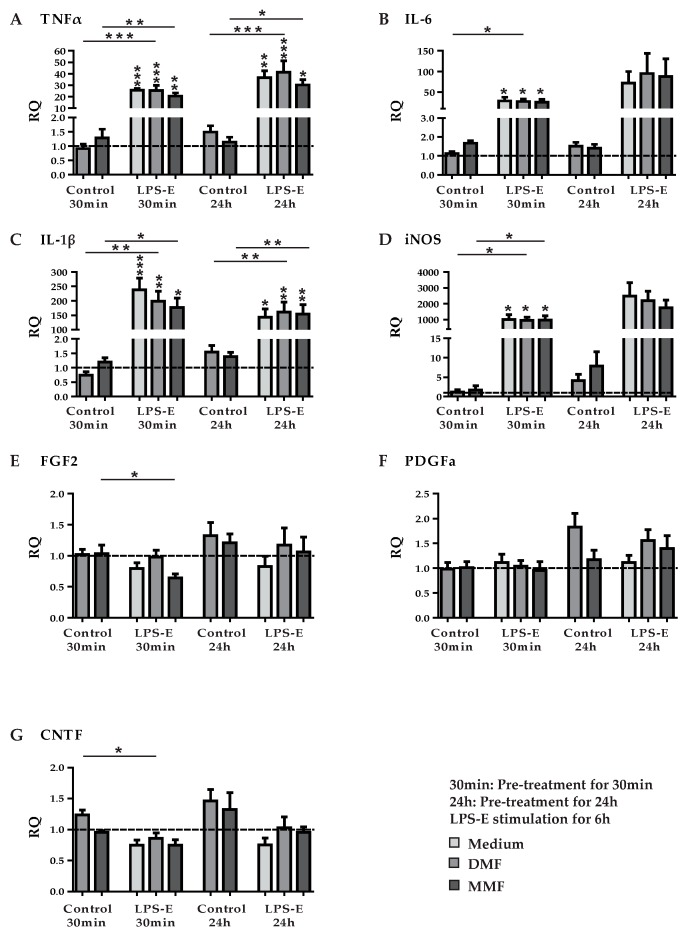
Gene expression of pro-inflammatory cytokines and growth factors in astrocytes pre-treated with DMF or MMF and stimulated with LPS. Astrocytes were pre-treated with medium, 10 µM DMF or 10 µM MMF for 30 min or 24 h and afterwards stimulated with 10 ng/mL LPS-E (lipopolysaccharide from *Escherichia coli* 055:B5). Graphs show mRNA expression fold changes of TNFα (**A**), IL-6 (**B**), IL-1β (**C**), iNOS (**D**), FGF2 (**E**), PDGFa (**F**) and CNTF (**G**) compared to the control group (astrocytes only treated with medium) and normalized with HPRT-1 using the ΔΔCT method. Data are presented as the arithmetic means ± SEM of *n* = 4. Significant differences are marked by asterisks (* *p* < 0.05; ** *p* < 0.01; *** *p* < 0.001).

**Table 1 brainsci-09-00241-t001:** Primer used for real-time quantitative polymerase chain reaction (qPCR).

Gene	Gene Expression Assay Number
*NGF*	Rn_01533872_m1
*BNDF*	Rn_00560868_m1
*GDNF*	Rn_00569510_m1
*PDGFa*	Rn_00709363_m1
*FGF2*	Rn_00570809_m1
*CNTF*	Rn_00755092_m1
*IL-1* *β*	Rn_00580432_m1
*IGF-1*	Rn_00710306_m1
*TNF* *α*	Rn_99999017_m1
*iNOS*	Rn_00561646_m1
*IL-6*	Rn_01410330_m1
*HPRT*	Rn_01527840_m1

*NGF* nerve growth factor, *BDNF* brain-derived neurotrophic factor, *GDNF* glial cell-derived neurotrophic factor, *PDGFa* platelet-derived growth factor subunit A, *FGF2* fibroblast growth factor, *CNTF* ciliary neurotrophic factor, *IL-1β* interleukin 1 beta, *IGF-1* insulin-like growth factor 1, *TNFα* tumor necrosis factor alpha, *iNOS* inducible nitric oxide synthase, IL-6 interleukin 6, *HPRT-1* hypoxanthine-guanine-phosphoribosyl-transferase 1.

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
