# Peer review of "Fumaric Acids Do Not Directly Influence Gene Expression of Neuroprotective Factors in Highly Purified Rodent Astrocytes"

_brainsci, 2019, doi:10.3390/brainsci9090241_

Round 1

Reviewer 1 Report

Dear Editor

The manuscript by Pars and Brieskorn et al reports that the neuroprotective effect DMF and its active metabolite MMF isn’t mediated by their direct effect on gene expression in stimulated astrocytes. The manuscript is well written and the study is based on their previous work investigating the same compounds on rodent microglia. The design of the study and the technical quality of the work appear convincing and results can be of general interest.

However, there is a number of major and minor points that would need to be addressed in order to improve the quality of this paper before it can be considered for publication:

Major points:

-MS is a chronic disorder and inflammation is a landmark for this disease. The design of the study includes pre-treatment with DMF and MMF before inducing the insult to the cells in vitro, this isn’t more pathologically relevant than administering them following the cellular insult since these medicines aren’t mainly used as prophylactic agents. Despite the authors have included hints trying to address this limitation but this isn’t enough. Both set of data are required to be included in the revised version of the manuscript.

- Authors have investigated a good number of genes but surprisingly no single protein. The discrepancies between gene and protein data is a well-known fact and since the aim of this study was to provide a mechanistic insight, authors need to investigate two or more proteins to solidify the outcomes of the suggested effect.

-Authors need to justify not opting for human primary astrocytes since this will be a more physiologically relevant model than rodent astrocytes.

-The aim of this study was to provide a mechanistic insight; however, there was a lack of an in-depth discussion. The second and third paragraphs were repetitive from the results with no new added information. Moreover, authors should discuss their data in the light of other cellular insults such as hypoxia since it can exacerbate inflammation and induced a number of signaling cascades. Moreover, inflammation can also trigger hypoxia by damaging mitochondria and endothelial cells to impair blood flow regulation. References to be included:

https://www.ncbi.nlm.nih.gov/pubmed/30052113

https://www.ncbi.nlm.nih.gov/pubmed/29311824

https://www.ncbi.nlm.nih.gov/pubmed/12559509

Minor points:

- A general introduction about astrocytes and their role should be included in the introduction. This intro should be followed by a review reference from outside the neurodegeneration field because this is important to understand significance of the study. A suggested reference:

https://www.ncbi.nlm.nih.gov/pubmed/21075920

-In the materials and methods, a statement about the mycoplasma testing should be included.

-Line 47: add a reference after “used in psoriasis treatment for many years” should be added.

-Line 64: typo- we demonstrated.

-Line 153: change “stimulation with medium” to “experimental control”.

-Line 220: “there was no regulation of BDNF and FGF2”, this is a confusing. Avoid short sentences and authors could rephrase to something like “there was no changes in the expression profile”.

Best regards.

Author Response

We would like to thank Reviewer 1 for the feedback and recommendations which helped us to improve our work. 

For the author's reply please see the attachment.  

Response to Reviewer 1 Comments

Major points

Point 1: MS is a chronic disorder and inflammation is a landmark for this disease. The design of the study includes pre-treatment with DMF and MMF before inducing the insult to the cells in vitro, this isn’t more pathologically relevant than administering them following the cellular insult since these medicines aren’t mainly used as prophylactic agents. Despite the authors have included hints trying to address this limitation but this isn’t enough. Both set of data are required to be included in the revised version of the manuscript. 

Response 1: It is correct that MS is known as a chronic disorder and inflammation is a hallmark of this disease. The goal of all immunomodulatory treatments including DMF is to prevent the development of new lesions. DMF is not a relapse treatment where the drug is added after the acute inflammation has started. Thus, DMF is used as a prophylactic therapy to influence inflammation and that is why we set up the experiments using DMF as pre-treatment similar to the everyday use as a drug (TecfideraÒ). The scenario where there is no acute inflammation is reflected by the first set of experiments (Fig. 1a) where non-stimulated astrocytes have been treated with DMF and MMF.

Point 2: Authors have investigated a good number of genes but surprisingly no single protein. The discrepancies between gene and protein data is a well-known fact and since the aim of this study was to provide a mechanistic insight, authors need to investigate two or more proteins to solidify the outcomes of the suggested effect.

Response 2: It is correct that a discrepancy in gene and protein level regulation has been reported before, and there are publications discussing the issue that a high gene expression is not correlated to corresponding protein levels (Maier, T.; Guell, M.; Serrano, L. Correlation of mRNA and protein in complex biological samples. FEBS Lett. 2009, 583, 3966–3973. [CrossRef]. De Sousa Abreu, R.; Penalva, L.O.; Marcotte, E.M.; Vogel, C. Global signatures of protein and mRNA expression levels. Mol. Biosyst. 2009, 5, 1512–1526. [CrossRef] [PubMed]). However, we could not demonstrate any influence of DMF or MMF on the mRNA level. Thus, we do not expect a discrepancy on the protein level.

Point 3: Authors need to justify not opting for human primary astrocytes since this will be a more physiologically relevant model than rodent astrocytes.

Response 3: We completely agree. Previous publications described that rodent astrocytes react in a different way to toll-like receptor ligands or cytokines compared to human astrocytes (Glia. 2014 June ; 62(6): 999–1013. doi:10.1002/glia.22657). We wanted to investigate a possible effect of DMF and MMF on certain gene expression in astrocytes to the same inflammatory insult. The aim of this study was not to investigate a possible difference between rodent or human astrocytes. However, previous studies have described concordant effects of DMF and MMF on both human and rodent astrocytes (Galloway DA, Williams JB, Moore CS. Ann Clin Transl Neurol. 2017 May 4;4(6):381-391. doi: 10.1002/acn3.414. eCollection 2017 Jun). 

Point 4: The aim of this study was to provide a mechanistic insight; however, there was a lack of an in-depth discussion. The second and third paragraphs were repetitive from the results with no new added information. Moreover, authors should discuss their data in the light of other cellular insults such as hypoxia since it can exacerbate inflammation and induced a number of signaling cascades. Moreover, inflammation can also trigger hypoxia by damaging mitochondria and endothelial cells to impair blood flow regulation. References to be included:

https://www.ncbi.nlm.nih.gov/pubmed/30052113

https://www.ncbi.nlm.nih.gov/pubmed/29311824

https://www.ncbi.nlm.nih.gov/pubmed/12559509

Response 4: Thank you for the recommendation. We completed the discussion, including the suggested references.

Minor Points:

Point 5: A general introduction about astrocytes and their role should be included in the introduction. This intro should be followed by a review reference from outside the neurodegeneration field because this is important to understand significance of the study. A suggested reference:

https://www.ncbi.nlm.nih.gov/pubmed/21075920

Response 5: Thank you for the recommendation. We added a general introduction including the illustrative reference.

Point 6: In the materials and methods, a statement about the mycoplasma testing should be included.

Response 6: Thank you for the recommendation. In fact there were no microscopical abnormalities nor suspicious changes of the cell cycles of the cell cultures so we did not assume a mycoplasma testing as indicated.

Point 7: Line 47: add a reference after “used in psoriasis treatment for many years” should be added.

Response 7: Thank you for the recommendation. An appropriate reference was added to the information.

Point 8: Line 64: typo- we demonstrated.

Response 8: Thank you for the note. We changed the expression.

Point 9: Line 153: change “stimulation with medium” to “experimental control”.

Response 9: Thank you for the note. We changed the sentence as suggested.

Point 10: Line 220: “there was no regulation of BDNF and FGF2”, this is a confusing. Avoid short sentences and authors could rephrase to something like “there was no changes in the expression profile”.

Response 10: Thank for the advice. We changed the sentence as suggested.

Reviewer 2 Report

The manuscript by Pars et al describes the effects of dimethylfumarate (DMF) and its metabolite, monomethylfumarate (MMF), on highly purified primary astrocytes in physiological conditions and in pro-inflammatory conditions induced by LPS. The effects on gene expression of selected growth factors and cytokines were assessed by qPCR. The findings are adding an important in vitro data to already exciting knowledge, which might help to elucidate the anti-inflammatory effects of DMF in such pathological setting as multiple sclerosis. The strength of the paper is an employing a protocol of primary astrocyte purification yielding in almost pure astrocyte culture, thus the obtained results are “clean” from the effects of other type of potentially co-isolated cells.

Minor points:

Line 142: “..had no effect on unstimulated cells compared to control”- it is not clear. The control is cells treated with media alone, so what are “unstimulated” cells. Perhaps it would be better to call them “treated cells”.

Line 255: Nrf2 is first time mentioned without explanation what it is.

Figures: The significant marks in Figures, like in Fig. 1b, 2, and 3 are too small. Please enlarge the font size.

Author Response

We would like to thank Reviewer 2 for the feedback and the recommendations which helped us to improve our work. 

For author's reply please see the attachment.  

Response to Reviewer 2 Comments

Minor points

Point 1: Line 142: “..had no effect on unstimulated cells compared to control”- it is not clear. The control is cells treated with media alone, so what are “unstimulated” cells. Perhaps it would be better to call them “treated cells”.

Response 1: Thank you for the recommendation. We changed the sentence as suggested.

Point 2: Line 255: Nrf2 is first time mentioned without explanation what it is.

Response 2: We wrote out nuclear factor E2-related factor 2 (Nrf2).

Point 3: Figures: The significant marks in Figures, like in Fig. 1b, 2, and 3 are too small. Please enlarge the font size.

Response 3: Thank you for the recommendation. We changed the font size of all significant marks in all figures.

Round 2

Reviewer 1 Report

Dear Editor

I would like to thank the authors for their efforts on trying to address my comments and concerns.

Some of the comments are convincing but there is a still number of points that would need to be accordingly addressed before the manuscript can be accepted. It is important that the authors focus on addressing the raised issues rather than attempting to divert the reviewers’ and editor’s attention away from the fact that the review comments were not fully addressed such as in point 4.

Please find attached my detailed comments.

Best.

Major points

Point 1: MS is a chronic disorder and inflammation is a landmark for this disease. The design of the study includes pre-treatment with DMF and MMF before inducing the insult to the cells in vitro, this isn’t more pathologically relevant than administering them following the cellular insult since these medicines aren’t mainly used as prophylactic agents. Despite the authors have included hints trying to address this limitation but this isn’t enough. Both set of data are required to be included in the revised version of the manuscript. 

Response 1: It is correct that MS is known as a chronic disorder and inflammation is a hallmark of this disease. The goal of all immunomodulatory treatments including DMF is to prevent the development of new lesions. DMF is not a relapse treatment where the drug is added after the acute inflammation has started. Thus, DMF is used as a prophylactic therapy to influence inflammation and that is why we set up the experiments using DMF as pre-treatment similar to the everyday use as a drug (TecfideraÒ). The scenario where there is no acute inflammation is reflected by the first set of experiments (Fig. 1a) where non-stimulated astrocytes have been treated with DMF and MMF.

Comment: Thanks for clarifying it. This paragraph needs to go into the experimental design so it can be easier to follow.

Point 2: Authors have investigated a good number of genes but surprisingly no single protein. The discrepancies between gene and protein data is a well-known fact and since the aim of this study was to provide a mechanistic insight, authors need to investigate two or more proteins to solidify the outcomes of the suggested effect.

Response 2: It is correct that a discrepancy in gene and protein level regulation has been reported before, and there are publications discussing the issue that a high gene expression is not correlated to corresponding protein levels (Maier, T.; Guell, M.; Serrano, L. Correlation of mRNA and protein in complex biological samples. FEBS Lett. 2009, 583, 3966–3973. [CrossRef]. De Sousa Abreu, R.; Penalva, L.O.; Marcotte, E.M.; Vogel, C. Global signatures of protein and mRNA expression levels. Mol. Biosyst. 2009, 5, 1512–1526. [CrossRef] [PubMed]). However, we could not demonstrate any influence of DMF or MMF on the mRNA level. Thus, we do not expect a discrepancy on the protein level.

Comment: The fact that authors haven’t seen a change at the mRNA level isn’t a solid reason to exclude studying protein expression profile for a number of selected targets especially when the aim of the study is to provide a mechanistic insight. A paragraph at the discussion describing the limitations of the author’s work is encouraged.

Point 3: Authors need to justify not opting for human primary astrocytes since this will be a more physiologically relevant model than rodent astrocytes.

Response 3: We completely agree. Previous publications described that rodent astrocytes react in a different way to toll-like receptor ligands or cytokines compared to human astrocytes (Glia. 2014 June ; 62(6): 999–1013. doi:10.1002/glia.22657). We wanted to investigate a possible effect of DMF and MMF on certain gene expression in astrocytes to the same inflammatory insult. The aim of this study was not to investigate a possible difference between rodent or human astrocytes. However, previous studies have described concordant effects of DMF and MMF on both human and rodent astrocytes (Galloway DA, Williams JB, Moore CS. Ann Clin Transl Neurol. 2017 May 4;4(6):381-391. doi: 10.1002/acn3.414. eCollection 2017 Jun). 

Comment: I would like to thank the authors for nicely addressing this point. This can be added as one of the limitations for the current study or a future direction.

Point 4: The aim of this study was to provide a mechanistic insight; however, there was a lack of an in-depth discussion. The second and third paragraphs were repetitive from the results with no new added information. Moreover, authors should discuss their data in the light of other cellular insults such as hypoxia since it can exacerbate inflammation and induced a number of signaling cascades. Moreover, inflammation can also trigger hypoxia by damaging mitochondria and endothelial cells to impair blood flow regulation. References to be included:

https://www.ncbi.nlm.nih.gov/pubmed/30052113

https://www.ncbi.nlm.nih.gov/pubmed/29311824

https://www.ncbi.nlm.nih.gov/pubmed/12559509

Response 4: Thank you for the recommendation. We completed the discussion, including the suggested references.

Comment: Authors failed to address this point and the discussion needs to be improved.

For example, hypoxia is one of the major insults, authors need to introduce a paragraph about its role and refer to different cellular mechanisms that could underline its effect as indicated by the three suggested references or more. This can be applied to other cellular insults in order to enrich the discussion.

A sentence at the discussion describing the future directions is encouraged.

Minor Points: Point 5: A general introduction about astrocytes and their role should be included in the introduction. This intro should be followed by a review reference from outside the neurodegeneration field because this is important to understand significance of the study. A suggested reference:

https://www.ncbi.nlm.nih.gov/pubmed/21075920

Response 5: Thank you for the recommendation. We added a general introduction including the illustrative reference.

Comment: Thanks.

Point 6: In the materials and methods, a statement about the mycoplasma testing should be included.

Response 6: Thank you for the recommendation. In fact there were no microscopical abnormalities nor suspicious changes of the cell cycles of the cell cultures so we did not assume a mycoplasma testing as indicated.

Comment: Thanks for adding a statement in materials and methods.

Point 7: Line 47: add a reference after “used in psoriasis treatment for many years” should be added.

Response 7: Thank you for the recommendation. An appropriate reference was added to the information.

Comment: Thanks.

Point 8: Line 64: typo- we demonstrated.

Response 8: Thank you for the note. We changed the expression.

Comment: Thanks.

Point 9: Line 153: change “stimulation with medium” to “experimental control”.

Response 9: Thank you for the note. We changed the sentence as suggested.

Comment: Thanks.

Point 10: Line 220: “there was no regulation of BDNF and FGF2”, this is a confusing. Avoid short sentences and authors could rephrase to something like “there was no changes in the expression profile”.

Response 10: Thank for the advice. We changed the sentence as suggested.

Comment: Thanks.

Author Response

Dear Editor

We would like to thank the reviewer for the patient work and appreciate the feedback which allows us to improve our work. Much to our regret, apparently we could not address all of the raised issues properly in the first place. We really hope the new changes and completions will be convincing.

Please see the respective responses and the adapted revision.

Kind regards.  

Dear Editor

I would like to thank the authors for their efforts on trying to address my comments and concerns.

Some of the comments are convincing but there is a still number of points that would need to be accordingly addressed before the manuscript can be accepted. It is important that the authors focus on addressing the raised issues rather than attempting to divert the reviewers’ and editor’s attention away from the fact that the review comments were not fully addressed such as in point 4.

Please see my comments below.

Best.

Dear Editor

We would like to thank the reviewer for the patient work and appreciate the feedback which allows us to improve our work. Much to our regret, apparently we could not address all of the raised issues properly in the first place. We really hope the new changes and completions will be convincing.

Please see the respective responses and the adapted revision.

Kind regards. 

Major points

Point 1: MS is a chronic disorder and inflammation is a landmark for this disease. The design of the study includes pre-treatment with DMF and MMF before inducing the insult to the cells in vitro, this isn’t more pathologically relevant than administering them following the cellular insult since these medicines aren’t mainly used as prophylactic agents. Despite the authors have included hints trying to address this limitation but this isn’t enough. Both set of data are required to be included in the revised version of the manuscript. 

Response 1: It is correct that MS is known as a chronic disorder and inflammation is a hallmark of this disease. The goal of all immunomodulatory treatments including DMF is to prevent the development of new lesions. DMF is not a relapse treatment where the drug is added after the acute inflammation has started. Thus, DMF is used as a prophylactic therapy to influence inflammation and that is why we set up the experiments using DMF as pre-treatment similar to the everyday use as a drug (TecfideraÒ). The scenario where there is no acute inflammation is reflected by the first set of experiments (Fig. 1a) where non-stimulated astrocytes have been treated with DMF and MMF.

Comment: Thanks for clarifying it. This paragraph needs to go into the experimental design so it can be easier to follow.

Response: Thank you for the note. The expermental design was adapted.

Point 2: Authors have investigated a good number of genes but surprisingly no single protein. The discrepancies between gene and protein data is a well-known fact and since the aim of this study was to provide a mechanistic insight, authors need to investigate two or more proteins to solidify the outcomes of the suggested effect.

Response 2: It is correct that a discrepancy in gene and protein level regulation has been reported before, and there are publications discussing the issue that a high gene expression is not correlated to corresponding protein levels (Maier, T.; Guell, M.; Serrano, L. Correlation of mRNA and protein in complex biological samples. FEBS Lett. 2009, 583, 3966–3973. [CrossRef]. De Sousa Abreu, R.; Penalva, L.O.; Marcotte, E.M.; Vogel, C. Global signatures of protein and mRNA expression levels. Mol. Biosyst. 2009, 5, 1512–1526. [CrossRef] [PubMed]). However, we could not demonstrate any influence of DMF or MMF on the mRNA level. Thus, we do not expect a discrepancy on the protein level.

Comment: The fact that authors haven’t seen a change at the mRNA level isn’t a solid reason to exclude studying protein expression profile for a number of selected targets especially when the aim of the study is to provide a mechanistic insight. A paragraph at the discussion describing the limitations of the author’s work is encouraged.

Response: Thank you for the note. A new section “Limitations of the work” was added, and this point is one of the limitations mentioned there.

Point 3: Authors need to justify not opting for human primary astrocytes since this will be a more physiologically relevant model than rodent astrocytes.

Response 3: We completely agree. Previous publications described that rodent astrocytes react in a different way to toll-like receptor ligands or cytokines compared to human astrocytes (Glia. 2014 June ; 62(6): 999–1013. doi:10.1002/glia.22657). We wanted to investigate a possible effect of DMF and MMF on certain gene expression in astrocytes to the same inflammatory insult. The aim of this study was not to investigate a possible difference between rodent or human astrocytes. However, previous studies have described concordant effects of DMF and MMF on both human and rodent astrocytes (Galloway DA, Williams JB, Moore CS. Ann Clin Transl Neurol. 2017 May 4;4(6):381-391. doi: 10.1002/acn3.414. eCollection 2017 Jun). 

Comment: I would like to thank the authors for nicely addressing this point. This can be added as one of the limitations for the current study or a future direction.

Response: Thank you for the feedback. A new section “Limitations of the work” was added, and this point is one of the limitations mentioned there.

Point 4: The aim of this study was to provide a mechanistic insight; however, there was a lack of an in-depth discussion. The second and third paragraphs were repetitive from the results with no new added information. Moreover, authors should discuss their data in the light of other cellular insults such as hypoxia since it can exacerbate inflammation and induced a number of signaling cascades. Moreover, inflammation can also trigger hypoxia by damaging mitochondria and endothelial cells to impair blood flow regulation. References to be included:

https://www.ncbi.nlm.nih.gov/pubmed/30052113

https://www.ncbi.nlm.nih.gov/pubmed/29311824

https://www.ncbi.nlm.nih.gov/pubmed/12559509

Response 4: Thank you for the recommendation. We completed the discussion, including the suggested references.

Comment: Authors failed to address this point and the discussion needs to be improved.

For example, hypoxia is one of the major insults, authors need to introduce a paragraph about its role and refer to different cellular mechanisms that could underline its effect as indicated by the three suggested references or more. This can be applied to other cellular insults in order to enrich the discussion.

A sentence at the discussion describing the future directions is encouraged.

Response: Thank you for the feedback. We hope we could address this point as suggested to improve our work. We included the different patterns of MS lesions (pattern I, pattern II, and pattern III) as described by Lassmann and colleagues (Lassmann H, Bruck W, Lucchinetti CF. The immunopathology of multiple sclerosis: an overview. Brain Pathol 2007;17(2):210-218.). We mentioned the important role of hypoxia and referred to the production of oxygen and nitric oxide radials. We also tried to point out the cellular mechanisms, which are demonstrated as links between hypoxia and inflammation (the prolylhydroxylase (PHD) related pathways, hypoxia inducible factor 1 alpha (HIF-1a), nuclear factor kappa B (NF-kB)). We added a sentence to encourage further research to elucidate the mode of action of DMF as MS therapy.

Minor Points: Point 5: A general introduction about astrocytes and their role should be included in the introduction. This intro should be followed by a review reference from outside the neurodegeneration field because this is important to understand significance of the study. A suggested reference:

https://www.ncbi.nlm.nih.gov/pubmed/21075920

Response 5: Thank you for the recommendation. We added a general introduction including the illustrative reference.

Comment: Thanks.

Point 6: In the materials and methods, a statement about the mycoplasma testing should be included.

Response 6: Thank you for the recommendation. In fact there were no microscopical abnormalities nor suspicious changes of the cell cycles of the cell cultures so we did not assume a mycoplasma testing as indicated.

Comment: Thanks for adding a statement in materials and methods.

Point 7: Line 47: add a reference after “used in psoriasis treatment for many years” should be added.

Response 7: Thank you for the recommendation. An appropriate reference was added to the information.

Comment: Thanks.

Point 8: Line 64: typo- we demonstrated.

Response 8: Thank you for the note. We changed the expression.

Comment: Thanks.

Point 9: Line 153: change “stimulation with medium” to “experimental control”.

Response 9: Thank you for the note. We changed the sentence as suggested.

Comment: Thanks.

Point 10: Line 220: “there was no regulation of BDNF and FGF2”, this is a confusing. Avoid short sentences and authors could rephrase to something like “there was no changes in the expression profile”.

Response 10: Thank for the advice. We changed the sentence as suggested.

Comment: Thanks.